# A Blinded Investigation: Accentuated NK Lymphocyte CD335 (NKp46) Expression Predicts Pregnancy Failures

**DOI:** 10.3390/diagnostics13111845

**Published:** 2023-05-25

**Authors:** Boris V. Dons’koi, Serhiy M. Baksheev, Irina O. Sudoma, Ihor E. Palyha, Ksenia G. Khazhylenko, Dariia V. Zabara, Yaroslava I. Anoshko, Viktor E. Dosenko, Evgen I. Dubrovsky

**Affiliations:** 1Laboratory of Immunology, Institute of Pediatrics, Obstetrics and Gynecology Named after Academician O. Lukyanova of the National Academy of Medical Sciences of Ukraine, Mayborody Str. 8, 04050 Kyiv, Ukraine; dariia.osypchuk@gmail.com (D.V.Z.); yasia-anoshko@ukr.net (Y.I.A.); udjin1785@gmail.com (E.I.D.); 2Private Enterprise “First Social Medical Laboratory “ESCULAB””, 79010 Lviv, Ukraine; 33th Maternity Hospital, 03148 Kyiv, Ukraine; s@baksheev.com.ua; 4NADIYA Clinic, 03037 Kyiv, Ukraine; i.sudoma@ivf.com.ua; 5LELEKA Maternity Hospital, 04075 Kyiv, Ukraine; 6Klinika Reproduktsiyi Lyudyny “Al’ternatyva”, 79041 Lviv, Ukraine; ivf.lviv@gmail.com; 7“RID” Fertility Centre, 02000 Kyiv, Ukraine; 8Department of General and Molecular Pathophysiology, Bogomoletz Institute of Physiology, National Academy of Sciences of Ukraine, 02000 Kyiv, Ukraine; dosenko@biph.kiev.ua

**Keywords:** NK lymphocytes, CD335, NKp46, pregnancy failures

## Abstract

Aim: NKp46 is an NK cell receptor uniquely expressed by NK cells and a small subset of innate lymphoid cells. In our previous studies, we suggested a tight connection between the activity of NK cells and the expression of NKp46 and supported the clinical significance of NKp46 expression in NK cells in women with reproductive failures. In this study, we investigated the expression of NKp46 in NK cells in the peripheral blood of women in early pregnancy and analyzed its association with pregnancy loss. Methods: In a blinded study, we examined blood samples and analyzed the subsequent pregnancy outcomes from 98 early pregnant women (5th–7th week of gestation—w.g.) and 66 women in the 11th–13th week of pregnancy who served as controls. We studied the expression of NKp46 and the levels of anti-cardiolipin antibodies (aCL). The results of aCL were shared with the clinic, while the expression of NKp46 was blinded and not analyzed until the end of the study. Results: A misbalance in the NKp46^+^NK cells subpopulations was associated with an unfavorable ongoing pregnancy. A decreased level of NKp46^high^ cells (<14%) was strongly associated with miscarriage. A decreased level of the double-bright subpopulation (NKp46^hight^CD56^++^) also was a negative prognostic factor for the pregnancy course, but its increased level (>4%) was strongly associated with a successful pregnancy course. Conclusions: Our results showed that accentuated levels of NKp46^+^NK cells lead to a negative prognosis for early pregnancy courses in women.

## 1. Introduction

Natural killer (NK) cells are the predominant innate lymphocyte subset. Through their ability to mediate killing and to produce soluble factors, NK cells perform multitudes of immunological functions [1,2,3].

Natural killer cells play a crucial role in the successful reproductive process, and many studies have demonstrated their fundamental role in reproduction [4,5,6]. However, attempts to assess the qualitative and quantitative characteristics of NK cells and to interpret and implement these data still lead to conflicting conclusions [7,8,9].

One of the possible causes of the above are the different approaches used to evaluate and analyze the various parameters of NK cells [10,11]. The techniques used for measuring NK activity are hard to standardize, while the assessment of the number of NK cells is a technically simple procedure, but this parameter alone does not represent the functional potential and heterogeneity of the NK population.

NKp46, also known as natural cytotoxicity receptor 1 (NCR1), is expressed by NK cells and has a crucial role in the antitumor activity, antiviral activity and autoimmune setting of NK cells [12,13]. A growing number of studies suggest that accentuations of NKp46 expression are associated with reproductive failures, such as recurrent pregnancy failures and implantation failures [14,15,16].

Previously [11], we found that a fraction of NKp46^+^NK cells have a prognostic value for accentuated NK cytotoxicity status, both low and high. These results showed that NKp46 expression presents a “link” between NK cell frequency and their function and affords grounds for using the assessment of NKp46^+^NK cells as a responsive, simple, cheap and reliable method for NK cytotoxicity assessment. Our data also support the clinical significance of NKp46 expression in NK cells in women with recurrent implantation failures [15].

This study was a part of blinded multicenter investigation of the significance of NK p46 expression for reproduction. In the first part, we found that an accentuation of NKp46 expression is associated with subsequent embryo implantation failures and pregnancy failures after different types of IVF [17].

In this part, we studied the diagnostic prognostic value of NKp46 expression in NK cells for the subsequent pregnancy outcomes in naturally conceived women. 

## 2. Patients and Methods

### 2.1. Blind Research Design

Patients were tested for anticardiolipin (aCL) IgG antibody and NKp46 phenotype (1 October 2019–10 March 2020). The results of aCL antibodies were returned to the clinic, and patients with high levels (>20 GPL IgG phospholipid units per milliliter) were excluded from further analysis. One patient with a level of aCL that was >20 GPL and two patients with congenital malformations were excluded from the early pregnancy group. The NKp46 phenotype results were kept in the primary data files (CellQest BDBioscience) and were not analyzed until the end of the study and were kept blinded. The end of the study was the end of the pregnancy of the last patient admitted to the study. After the end of the clinical phase, the results of the NKp46 tests were analyzed and sent to medical institutions. In parallel, the results were sent to independent experts (Dosenko VE., Oshovsky VI. and Veselovsky VV). Similarly, the clinical results of ongoing pregnancies were sent to the laboratory and independent experts for analysis. All laboratory investigations were carried out (1 May 2019 until 10 March 2020) before the first confirmed SARS-CoV-2 case in Ukraine. All patients’ pregnancy failures occurred before the first wave of SARS-CoV-2 in Ukraine.

Pregnancy failures were calculated as a part of a failed pregnancy of a confirmed ongoing pregnancy. Successful-term pregnancies were calculated as a part of a live birth after 32 wg. All participants signed an informed consent form before being enrolled in the study (approved by the Biomedical Ethics Committee of the Institute of Pediatrics, Obstetrics and Gynecology, National Academy of Medical Sciences of Ukraine, No. 6, according to the Declaration of Helsinki).

### 2.2. Early Pregnancy Groups

Patients who were previously selected for study according to a doctor’s opinion had a high chance of a positive ongoing pregnancy and did not require additional research and treatment. Patients were <36 years old (y/o) and had no history of repeated reproductive losses (not more than 1 previous pregnancy failure), autoimmune, chronic or acute infection. Anamnestic and clinical data are shown in Table 1.

Peripheral blood samples (n = 100) were obtained at the 5th–7th week of gestation (wg) (ultrasound confirmed a single uterine pregnancy). Two patients were excluded from the investigation (one medical abortion due to fetal abnormalities, and one had high levels of aCL antibodies > 20 GPL with a subsequent pregnancy failure at 19 wg). In the investigated group (n = 98), 72 women had a live birth at term (32–41 wg)—successful pregnancy (SP) group, 20 patients subsequently had a pregnancy failure—pregnancy failure group (PF), among which 18 patients had a pregnancy failure in the 1st trimester and 2 patients had a pregnancy failure in the 2nd trimester (both with normal fetal karyotypes). Four patients had a preterm (29–32 wg) birth and were excluded from the analysis. We did not obtain information from 2 patients about their pregnancy outcome (Appendix A).

### 2.3. Control Group

We tested pregnant ladies who underwent an 11 + 0–13 + 6 combined screening (n = 187) examination with the aim of assessing their personal theoretical risks of possible chromosomal aneuploidies and possible later obstetrical complications such as a preeclampsia (PE) and fetal growth restriction (FGR). The above-mentioned risks were calculated by the ASTRAIA program based on the algorithm of the Fetal Medicine Foundation (FMF, London, UK), and biochemical tests were carried out using the laboratory equipment of the B.R.A.H.M.S. and Roche firms. Ultrasound examinations were carried out by operators with valid certificates from the FMF. 

Unfortunately, due to the war in Ukraine, we were only able to contact and obtain clinical information regarding 78 results of ongoing pregnancy. Sixty-six women formed the controlled-birth-at-term uncomplicated pregnancy group after 35 wg (Control UP). Five women had preterm birth at 30–34 wg and were analyzed separately. We excluded 7 women from the Control UP group with subsequent preeclampsia, and 2 women were excluded from the study due to pregnancy failures (Appendix A).

#### 2.3.1. Assessment of NKp46 Expression

We analyzed the CD335 phenotype as described previously [11,15]. A total of 5 mL of peripheral blood was collected and delivered to the laboratory no later than 24 h after the collection. To determine NKp46 expression, 100 μL samples of whole blood were stained by FITC-, PE- and PE-Cy5-conjugated monoclonal antibodies to CD3 (555339), CD335 (NKp46) (557991) and CD56 (555517) (BD Bioscience, San Jose, CA, USA), respectively. Lysed and washed samples were analyzed using a FACScan flow cytometer using the CellQuest software (BD Bioscience, San Jose, CA, USA). 

#### 2.3.2. Gating Strategy for NKp46^+^NK Cells

Baseline NKp46 expression in NK cells was assessed by flow cytometry as previously shown [17]. CD3negative CD56+ NK cells were gated from the total lymphocyte population—%NK (Appendix A). In all samples, CD56bright NK cells almost unanimously expressed high levels of NKp46. Three different CD56dim NK CD335 (NKp46) phenotypes were revealed—NKp46high, NKp46medium and NKp46neg predominance. 

The percentage of NKp46^+^NKcells (%NKp46^+^NK) among all the lymphocytes was determined as both the NKp46^high^ and NKp46^medium^ subsets (Appendix A). In parallel to manual gating, standard invariant gating for NKp46^high^, NKp46^medium^ and NKp46^neg^ enumeration was conducted (Appendix A).

### 2.4. Determination of NK Cytotoxicity

NK cell activity was measured as described previously [11]. Target K562 (NK-sensitive, HLA-negative and chronic myelogenous leukemia) cells were labeled with 5 μM of Cell Tracker™ Green CMFDA (5-chloromethylfluoresceindiacetate), (MolecularProbes, Eugene, OR, USA) and Calcein-AM (O,O′-diacetate tetrakis(acetoxymethyl) ester) (Sigma Aldrich) for 20 min at 37 °C in a humidified 5% CO_2_ incubator. The labeled cells were washed twice in phosphate-buffered saline (PBS), resuspended in RPMI1640 with 10% newborn calf serum (NBCS) and counted using Flow-CountTM Fluorospheres (Beckman Coulter, USA). The effector cells (isolated on a Histopaque-1077 density gradient (“Sigma”, USA))—peripheral blood mononuclear cells (PBMC)—were co-incubated at effector/target ratios of 30/1, 15/1 and 7.5/1 for 2.5 h at 37 °C in an atmosphere of 5% CO_2_ in air. After the incubation period, the cells were mixed with 10μL of propidium iodide (PI) solution with a concentration of 2 mg/mL (SIGMA) in PBS to stain dead cells. For each E/T ratio, the NK cytotoxicity was measured by analyzing 10,000 target cells/samples using a FACScan flow cytometer (BD Bioscience, San Jose, CA, USA) equipped with the CellQuest software. 

As for the concept of “normal cytotoxicity”, the reference values of NK cytotoxicity favorable for reproductive prognosis were accepted in accordance with our previous works [11,18].

Cytotoxic activities > 30% (E/T ratio 10/1) and <10% (E/T ratio 10/1) were considered high and low, respectively. Later, these levels were clinically confirmed in the prospective cohort study [18]. In addition, the same “normal cytotoxicity levels” were defined as the levels where no correlation between NK% and NK cytotoxicity was observed [11].

### 2.5. Anti-Cardiolipin Antibodies Measurement

An aCL ELISA was performed according to our previous publication [19] and was sensitive to β2GP1-independent and β2GP1-dependent aCLa. Polystyrene plates (polySorp, Nunc, Roskilde, Denmark) were covered with either 20 µL/well of 50 µg/mL cardiolipin (CL) (Sigma-Aldrich, Steinheim, Germany) in ethanol or ethanol alone (as a control well). After that, the wells were blocked by 1.0% bovine serum albumin (BSA) in 0.05 M of Trizma buffer pH-8.5. The serum samples were instilled in the wells for an hour after diluting to 1:50 in RPMI medium with 0.5% BSA and 0.01% Tween-20. The amount of bounded aCLAb was determined by incubation with horseradish peroxidase-conjugated goat anti-human IgG (λ chain specific) and a reaction of peroxidase with TMB. The optical density (OD) was read at λ = 450 nm using a Multiscan device (LabSystems, Finland). Values below 10 GPL for CL were considered as negative (>20 GPL were considered as positive) and were excluded from the investigation.

### 2.6. Statistical Analysis

The statistical analysis of the results was performed using Fisher’s exact test (T, unpaired, non-parametric, two-sided *p* value) and a 95% confidence interval using the approximation of Woolf in Stat version 3.0 for Windows Graph Pad Software Inc., (San Diego, CA, USA). (A two-sided *p* value < 0.05 was considered significant).

## 3. Results

### 3.1. Early Pregnancy Group

A high percentage of NKp46^+^NK cells was significantly associated with a pregnancy failure. A total of 20% (4 of 20) of the women with a PF had a high amount of NKp46^+^NK (more than 95% of NK cells), while only 1.3% of the women with an SP had such an amount (1/72). The amount of NKp46^+^NK (more than 95% of NK cells) was a negative prognostic factor for pregnancy success (OR = 17, *p* = 0.008). However, further analysis revealed that a low percentage of NKp46^+^NK cells (less than 55% of NK cells) was also more common in women with a PF (3/20, 15%) compared to the group with an SP (5/72, 6.9%), but the difference did not reach the level of significance (*p* = 0.08). However, taken together, both NKp46^+^NK accentuations (increase—more than 95% of NK cells, and decrease—less than 55% of NK cells) gave a highly reliable negative prognostic value for pregnancy success (*p* = 0.0063) (Table 2), Figure 1B. Thus, a balanced amount of NKp46^+^NK cells (>55% and <95%) was highly reliably favorable for indicating a successful pregnancy (Table 2), as shown in Figure 1B.

As described in the Section 2, three different NK CD335 (NKp46) phenotypes were identified, namely NKp46^high^, NKp46^dim^, and NKp46^neg^ cells. A low percentage of NKp46^high^ NK cells as determined using the constant-gate method (<7%) was more common in the group of women with a PF (40%, 8 out of 20) compared to those with an SP (12.5%, 9 out of 72) (OR = 4.67, *p* = 0.009) (Table 2), as shown in Figure 2B. In other words, the majority of patients who had a successful pregnancy outcome and those from the control group had p46++ levels that were higher than 7%, while only 60% of patients with a subsequent pregnancy failure had such levels (Table 2), Figure 2B.

The same statistical results were obtained using the manual gating of the NKp46^high^ subpopulation. However, in this case, levels of Nkp46^high^ cells of less than 16% were associated with a negative prognosis for a successful pregnancy. This can be explained by the fact that manual gating always leads to an increase in the level of minor populations, but it still left a similar statistical significance in this case. 

With respect to the different NKp46 phenotypes, we found that a predominance of NKp46^dim^ cells (>70% of NK cells), particularly in women with a PF, was a negative prognostic factor for implantation failure (OR = 7.3, *p* = 0.0062) compared with women from the SP group (Table 2). 

In addition, the level of subpopulations such as double-bright NK cells (CD56^high^NKp46^high^) was a prognostic factor for pregnancy outcome. In the group of women with a PF, a low level (<2.5%) of double-bright NK cells (CD56^++^NKp46^high^) was more common (7/20 35%) than in the group with a successful pregnancy (10/72 and 13%, odds ratio = 3.338. The two-sided *p* value was 0.0484) (Table 2). A high percentage (>4%) of double-bright NK cells (CD56^++^NKp46^high^) was a positive prognostic factor for a successful pregnancy. In the group of women with an SP, a high percentage (0.4%) of double-bright NK cells (CD56^++^NKp46^high^) was more common, with a value of 62.5% (45/72), compared with the group with a PF, with a value of 25% (5/20) (OR-5.1 *p* = 0.0046). 

Patients with a subsequent PF had significantly increased BMI compared to the SP women, but no one in both groups had a BMI of more than 31. The majority of the failures in the PF patients were at early stage, and, naturally, SP was investigated at significantly later terms of pregnancy (6.9 w/g) on average compared to (6.25 w.g.) the PF group (Table 1).

### 3.2. Control Population

A high percentage of NKp46^+^NK cells (>95%) was found in 3% (2/66) of the patients from the control group, and a low percentage (<55%) of NKp46^+^NK cells was also found in 3% (2/66) of the patients Figure 1B. A low percentage ofNKp46^high^NK cells (<7%) in the control group was found in 7.5% (5/66) of the patients, and these results were comparable to those from the group of women with an SP (Figure 2B). Decreased levels (<2.5%) of double-bright NK cells (CD56 ++NKp46^high^) in the control group were found in 10.6% (7/66) of the patients.

In Figure 1A and Figure 2A, we demonstrate the average levels of NKp46 expression and NKp46++ levels, and they were similar in all groups. This is another vivid example of the fact that calculating the average levels of everything related to NK cells is an incorrect approach.

### 3.3. NK Cytotoxicity

High levels of NK cytotoxicity are weakly associated with a miscarriage. Thus, among the patients with a PF, 33.3% (4/12) had increased NK cytotoxicity, while the same NK cytotoxicity occurred in only 6.9% (2/29) of the women with an SP. (OR = 6.750, one-sided *p* = 0.0452). Among the same patients, a percentage of NKp46^+^NK cells > 95% was negative prognostic factor for the pregnancy course. A total of 25% of the patients with a PF had such an accentuation, and none had it within the SP group (OR = 21.737. The two-sided *p* value was 0.0206).

## 4. Discussion

Previous works have demonstrated immunological features, particularly changes in NKp46 expression in NK cells, in women with recurrent implantation failures and recurrent spontaneous abortions [14,15,17]. These studies of such groups of women could not exclude the influence of repeated miscarriage on immunological parameters, particularly the expression of NKp46. Therefore, a causal relationship between repeated reproductive losses and the expression of NKp46 on NK cells had remained only as assumption. In this study, we investigated the immunological parameters in early pregnant women without a burdened history and previous repeated reproductive losses.

We found that both high and low levels of NKp46^+^NK cells were unfavorable prognostic factors for the course of pregnancy. A combination of these factors as one parameter—an accentuated level of NKp46^+^NK cells—indicated an even more statistically significant negative prognosis for the course of pregnancy. So, we can now be more confident that accentuated levels of NKp46^+^NK cells can be causally related to miscarriages.

Elevated peripheral blood (PB) NKp46 expression has been shown in women with recurrent implantation failures [18], whilst at the same time, endometrial NKp46+ cells have been shown to be markedly decreased in women with recurrent spontaneous abortion [20,21,22]. 

We believe that the following contradictory results can be explained by the heterogeneity of NKp46 expression in NK cells. At least three different subsets of NK cells (p46neg, p46dim and p46++) can be distinguished based on NKp46 expression [23,24,25]. Recent works have suggested that these subsets differ not only in terms of phenotypes but also in terms of functions [16,26]. It has been shown that decidual NKp46^bright^ cells are related to cytokine production, and in women with recurrent pregnancy loss with karyotypically normal pregnancies, a reduction in NKp46^bright^ cells is thought to cause abnormal cytokine production (NK1 shift), thus leading to a miscarriage [16,26].

In our work, accentuated levels of some NKp46^+^NK cell subpopulations were common for women with a subsequent pregnancy loss. Thus, a decreased NKp46^bright^ NK cell subpopulation (both constant and manually gating) was a negative prognostic factor for the pregnancy course.

A decreased level of the double-bright subpopulation (NKp46^high^CD56^bright^) was also a negative prognostic factor for the pregnancy course, but an increased level (>4%) was strongly associated with a successful pregnancy course. 

We also compared manual gating versus constant gating in different populations and found that the gating strategy influenced the results, but only in terms of the threshold at which a given subpopulation had an unfavorable prognosis and not in terms of the sense of the accentuation. Thus, for the manual gating of the NKp46bright subset, the significant adverse event rate was <16%, whereas for constant gating it was <7%. However, in both cases, the accentuation of this parameter was significantly unfavorable for the reproductive process. The manual gating of small populations led to its increase. Our results showed that having all the NKp46 subsets in balanced proportions formed a favorable background for the pregnancy outcome, while an accentuated level of one subset affected the balance and formed an unfavorable phenotype for reproduction. 

Accentuated levels of NK phenotypes and their association with reproductive complications have been demonstrated by us repeatedly [27,28,29,30]. However, such an imbalanced phenotype may be a different prognostic factor in other health conditions [31,32,33]. Thus, an accentuated imbalanced NK function forms a highly specific immunophenotype that may be favorable for a specific challenge (such as infection or malignancy) but unfavorable for adaptation to other challenges (such as embryo implantation and pregnancy).

Moreover, we have shown that, in a small group, accentuated levels of NKp46^+^NK cells and their subpopulations are more prognostic for determining the course of pregnancy compared to NK cytotoxicity in these patients.

In terms of NK testing, uterine NK (uNK) cells are often considered the most important because they are directly in the path of the implanting embryo. However, uNK cell testing is not possible during pregnancy [1]. However, in our work, we have shown that the study of NKp46^+^NK cells from peripheral blood is a reliable indicator for predicting pregnancy complications and can be used during pregnancy.

However, we demonstrated a lot of NKp46+ subpopulations where either their accentuation led to an unfavorable prognosis or their balanced levels led to a normal physiological prognosis. It was not entirely clear which populations’ accentuation (NKp46neg or NKp46+) affected the reproductive outcome because an increase in one automatically led to a decrease in the other. Misbalance results can lead to unfavorable conditions for pregnancy outcomes. Previously, we have shown that a similar misbalance in NK populations was also associated with an unfavorable health status during an Antarctic expedition [34]. Therefore, it is necessary to conduct a deeper and more detailed study of the physiological and phenotypic properties of different NKp46 populations. 

In this study, we specifically demonstrated the differences between the analysis of average levels and the analysis of individual distributions of values in clinical groups. We once again demonstrated that the comparison of average levels is meaningless in these cases. For more than 10 years, we have been trying to convey this with various examples. The accentuation of NK levels and cytotoxicity [11,18] and accentuated expressions of CD8 [29], CD69 [35], CD158a [28] and CD335 [15,17] in NK always have a bilateral character in patients with reproductive losses compared to healthy fertile groups.

**Limitation of study:** We planned to analyze 300 women with a physiological pregnancy, and we performed an immunological examination of them, but the ongoing war in Ukraine and the migration of women to friendly countries prevented us from collecting clinical information about the majority of cases. We were also limited in the actual study by the SARS-CoV-2 pandemic. We finished the immunological investigation before the start of the first wave in Ukraine, and any pregnancy failure was not the result of SARS-CoV-2. Another limitation—limitation for clinical use—was the complexity of applying the gates strategy. For most cases, the isolation of p46negative or p46++ is obvious and does not create problems, but for some samples it is problematic. So, the issue of standardization is important. We used the constant-gate method in parallel with the manual one and showed the same reliability. Normal/accentuated values will be different between the manual-gate method and the constant one, but both showed significant differences. Just as in our previous studies, the boundaries of the favorable corridor were different. Thus, for non-pregnant women, a p46 expression of more than 93% was unfavorable for implantation and carrying a pregnancy [17]. In the actual study, the limit of unaffordability regarding the expression of p46 was already 95%. 

Of course, we understand that we studied one single, albeit important, receptor in NK cells. There are several other receptors in NK cells, and in addition to NK, a lot of immune cells participate in the “reproductive orchestra”. Therefore, we did not investigate the function of NKT, regulatory T cells and dendritic cells, etc., but their influence on the outcome of pregnancy definitely took place [36,37,38]. Previously, we showed that isolated accentuation has little effect on the subsequent pregnancy outcome, while in association with other accentuations, this accentuated immunophenotype amplifies the adverse effect [27]. It is also likely that the favorable state of other links can neutralize the negative effect of isolated accentuation. So, for further advancement, it is necessary to carry out a complex multiparametric study of immune subsets.

In conclusion, in early pregnant women without a burdened history and previous repeated reproductive losses, accentuated levels of NKp46^+^NK cells led to a negative prognosis for the pregnancy course. It is suggested that NKp46 expression in NK cells can be causally related to miscarriages. 

## Figures and Tables

**Figure 1 diagnostics-13-01845-f001:**
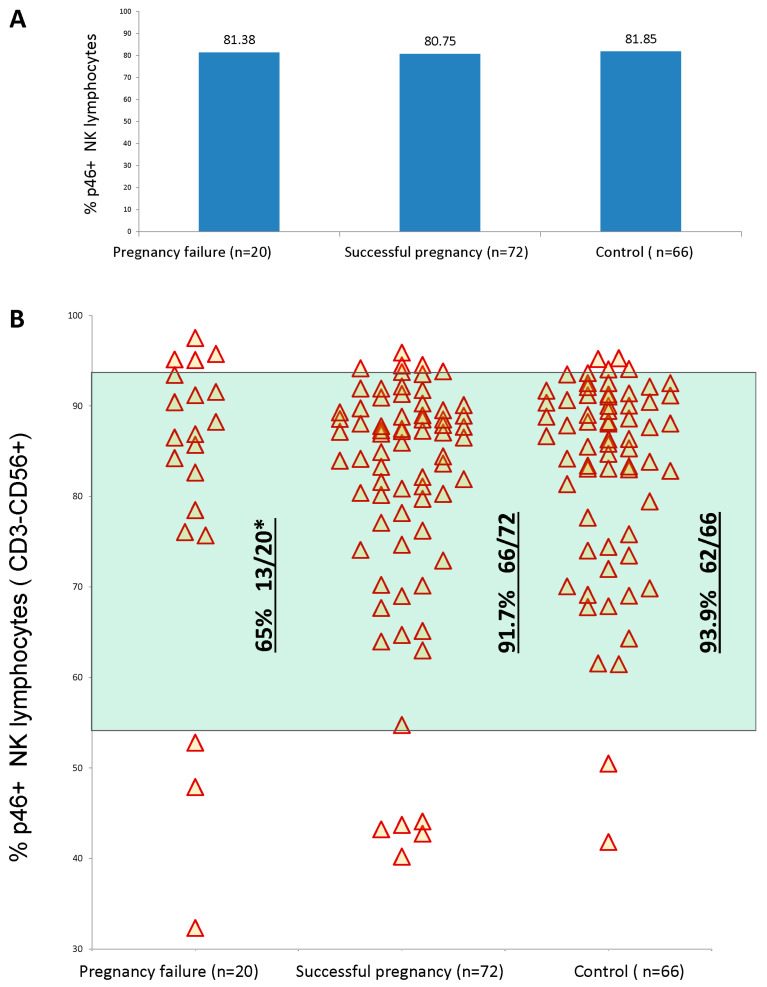
Comparison of NKp46 expression levels between patients with subsequent pregnancy failures or successful pregnancy outcomes and control group. (**A**) Average levels. (**B**) Individual level distributions. * Significant difference compared to successful pregnancy outcome (the two-sided *p* value was 0.0063, and the odds ratio = 5.923) and control groups (the two-sided *p* value was 0.0025, and the odds ratio = 8.34).

**Figure 2 diagnostics-13-01845-f002:**
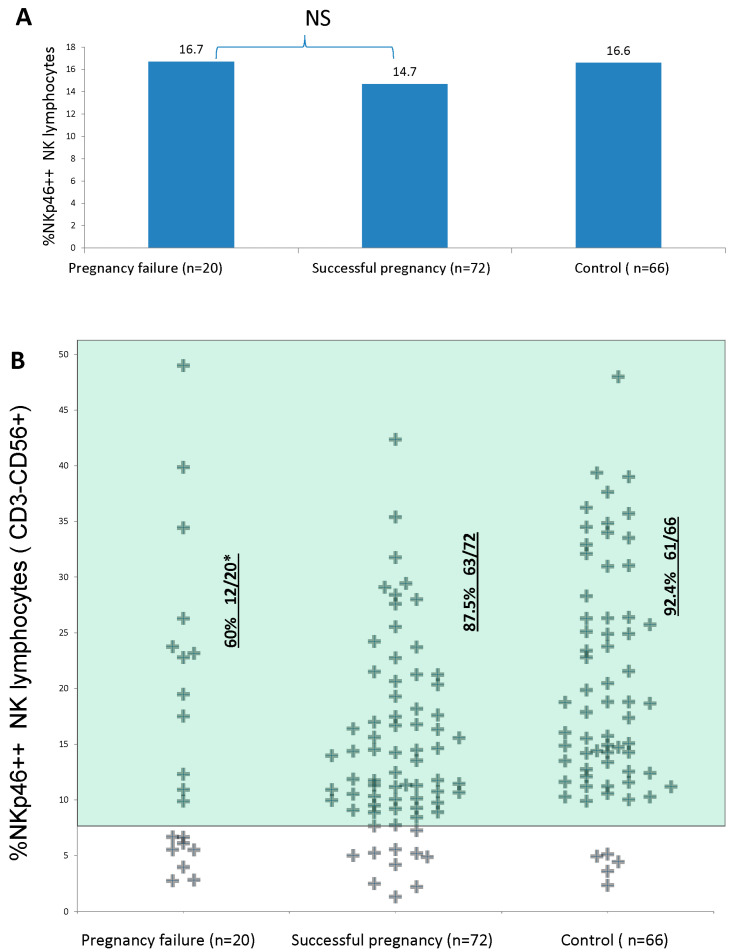
Comparison of NKp46bright levels between patients with subsequent pregnancy failures or successful pregnancy outcome and control group. (**A**) Average levels. (**B**) Individual level distributions. * Significant difference compared to successful pregnancy outcome (the two-sided *p* value was 0.0092, and the odds ratio = 4.667) and control groups (the two-sided *p* value was 0.0014, and the odds ratio = 8.13).

**Table 1 diagnostics-13-01845-t001:** Clinical and anamnestic parameters of investigated pregnancy groups.

	Pregnancy	Studied at (w.g.) (Average)	Age Years (Average)	Infertility in Anamnesis	Miscarriages in Anamnesis	Live Birth in Anamnesis	BMI (Average)
Early pregnant 5–8 w.g.	Pregnancy failures PF n = 20	6.25 ± 0.75 *	29.5 ± 3.6	15% (3/20)	25% (5/20)	40% (8/20)	24.3 ± 2.5 *
	Successful pregnancy (term LB > 35 w.g.) SP n = 72	6.9 ± 0.85	32.5 ± 5.2	2.8% (2/72)	18% (13/72)	34.7% (25/72)	21.7 ± 2.5

* Significantly different compared to SP group.

**Table 2 diagnostics-13-01845-t002:** Significance of balanced NKp46 phenotype and NKp46 subsets for pregnancy outcome. * According to the constant-gate method.

Investigated Groups	NK% Lymph. >18	%p46 Expression on NK (<55% or >95%)	% of p46dim Subset from NK (>70%)	% of p46++ Subset from NK (<7%) *	% of p46++ CD56++ Subset from NK <2.5%
Pregnancy failure PF n = 20	15% (3/20)	35% (7/20)	40% (8/20)	40% (8/20)	35% (7/20)
Successful pregnancy SP (Term LB > 32 w.g.) n = 72	18.5% (13/72)	8.3% (6/72)	16.6% (12/72)	12.5% (9/72)	13.8% (10/72)
Significance for PF compared to SP	ns	OR 5.9 *p* = 0.0063 95% CI 1.7–20.5	OR 3.3 *p* = 0.034 95% CI 1.03–9.8	OR 4.67 *p* = 0.009 95% CI 1.5–14.2	OR 3.3 *p* = 0.048 95% CI 1.07–10.4

## Data Availability

Not applicable.

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
