# Peer review of "A Blinded Investigation: Accentuated NK Lymphocyte CD335 (NKp46) Expression Predicts Pregnancy Failures"

_diagnostics, 2023, doi:10.3390/diagnostics13111845_

Round 1
Reviewer 1 Report
Please send the menuscript to an English native reviewer, for a thorough English review before resubmitting the study.
Author Response
Please send the menuscript to an English native reviewer, for a thorough English review before resubmitting the study.
-we have made detailed proofreading of the English language and made numerous corrections.
Reviewer 2 Report
Pregnancy failures are caused by chronic diseases of the mother such as high blood pressure, diabetes, thyroid diseases, or polycystic ovary syndrome (PCOS). These risks are related to the immune system, such as an autoimmune disorder. However, its underlying mechanism remains to be determined. The manuscript “A blinded investigation: accentuated NK lymphocyte CD335 (NKp46) expression predicts pregnancy failures” is devoted to giving information about the functions of pregnancy failures-related immune cells, especially, NK cells. Also, this manuscript’s results suggested that CD335 expression in NK cells may be related to pregnancy failures. Although this manuscript has good results, the authors suggest their theory using only two tables.
Especially, additional figures need to make it easier for the reader to read.
Some points have to be corrected.
Major points
1. The authors only mentioned pregnancy failure-related cells. Is there any possibility that other cells also are related to pregnancy failures? For example, NK T cells or regulatory T cells.
2. In this study, the authors focused on the CD335 expression of NK cells as a potential prediction of pregnancy failures. It is better to add a schematic model of their relationship and how they are related to pregnancy failures.
Minor points
1. Line 18: Add a comma after “studies”.
2. Line 95: Amend “was” to “were”.
3. Line 102: Check the sentence “We not obtained”. Is it right?
4. Line 229: Add a comma “In fig 1a and fig 2a”.
5. Line 234: Add “are” before “weakly”.
Author Response
Comments and Suggestions for Authors
Pregnancy failures are caused by chronic diseases of the mother such as high blood pressure, diabetes, thyroid diseases, or polycystic ovary syndrome (PCOS). These risks are related to the immune system, such as an autoimmune disorder. However, its underlying mechanism remains to be determined. The manuscript “A blinded investigation: accentuated NK lymphocyte CD335 (NKp46) expression predicts pregnancy failures” is devoted to giving information about the functions of pregnancy failures-related immune cells, especially, NK cells. Also, this manuscript’s results suggested that CD335 expression in NK cells may be related to pregnancy failures. Although this manuscript has good results, the authors suggest their theory using only two tables.
Especially, additional figures need to make it easier for the reader to read.
Some points have to be corrected.
Major points
- The authors only mentioned pregnancy failure-related cells. Is there any possibility that other cells also are related to pregnancy failures? For example, NK T cells or regulatory T cells.
We add it in limitation.
- In this study, the authors focused on the CD335 expression of NK cells as a potential prediction of pregnancy failures. It is better to add a schematic model of their relationship and how they are related to pregnancy failures.
In the current study, we obtained an interesting result regarding the clinical value of p46, but we still do not have the courage to build theoretical models of exactly how CD335 on NK affects reproduction... in our opinion, it is still too early for such models.
The only thing we constantly see is that it is better to have balanced levels of anything than accented ones.
“that too much is not healthy” "Too good isn't good, too" ))
Minor points
Line 18: Add a comma after “studies”.
-We correct it
- Line 95: Amend “was” to “were”.
We correct it
- Line 102: Check the sentence “We not obtained”. Is it right?
We correct it
- Line 229: Add a comma “In fig 1a and fig 2a”.
-We correct it
- Line 234: Add “are” before “weakly”.
-We correct it
Thank you for your attention to our article, advice and comments. We corrected the manuscript according to the comments. A detailed correction of the English language was also carried out.
Reviewer 3 Report
This is an interesting and well-conducted prospective study though at a difficult time for the country of the authors (war and SarsCov2 pandemic). Methods and statistical analysis is valid and conclusions are supported by the results. Discussion and review of literature are adequate. English language is satisfactory. I think the article should be published as it adds significact scientific information to current knowledge.
Author Response
Comments and Suggestions for Authors
This is an interesting and well-conducted prospective study though at a difficult time for the country of the authors (war and SarsCov2 pandemic). Methods and statistical analysis is valid and conclusions are supported by the results. Discussion and review of literature are adequate. English language is satisfactory. I think the article should be published as it adds significact scientific information to current knowledge.
-Thank you for the positive evaluation of our work.
-We have made detailed proofreading of the English language and made numerous corrections.
Reviewer 4 Report
Overall, the novelty, concept, manuscript writing of this study are all good enough for publication. This paper could be accepted for publication.
Author Response
Overall, the novelty, concept, manuscript writing of this study are all good enough for publication. This paper could be accepted for publication.
-Thank you for the positive evaluation of our work.
Reviewer 5 Report
Dear author’s
I was pleased to review your article and i have the following comment’s:
The subject is interesting and the main question of the study is very we’ll choose. Early pregnancy loss is a real problem in infertile couples undergoing ivf.
Please explain the reason that the control group was established as 11 -13 ws gestation?
The sample of the study is relatively small and further studies are necessarily in order to conclude that elevated levels of NKp46+NK cells lead to negative prognosis pregnancy course.
Please explain the novelty of the study and highlight the new information that your study results brings to the field.
A flowchart with the patients in the methodology section are useful.
In discussion section your result should be compared with the existing literature.
Minor English edis are necessary.
Author Response
Dear author’s
I was pleased to review your article and i have the following comment’s:
The subject is interesting and the main question of the study is very we’ll choose. Early pregnancy loss is a real problem in infertile couples undergoing ivf.
Please explain the reason that the control group was established as 11 -13 ws gestation?
-We agree that this additional control group looks like crap. However, this is all we managed to collect from an interesting project. We had a plan to study 300 women who are screening for the possibility of preeclampsia and look not only at the frequency of live births, but also at the condition of the child at the age of 1 year. However, after the start of the full-scale war, we of 200 women found reliable clinical information. only a small part. I hope that everyone else in European countries is okay with them. But we do not have the opportunity connect with them. So there were only 66 women who gave birth on time. This is a kind of additional control. Their p46 levels are concordant with women with successful pregnancies from the study group. So we kept these results because they increase the reliability of the results we obtained in the studied group.
The sample of the study is relatively small and further studies are necessarily in order to conclude that elevated levels of NKp46+NK cells lead to negative prognosis pregnancy course.
Please explain the novelty of the study and highlight the new information that your study results brings to the field.
It s a first investigation that made as prognostic blinded study format and especial important that its made on natural pregnant women without RPL anamnesis.
A flowchart with the patients in the methodology section are useful.
-We add supplementary tab1
In discussion section your result should be compared with the existing literature.
We add it
Minor English edis are necessary.
Thank you for your attention to our article, advice and comments. We corrected the manuscript according to the comments. A detailed correction of the English language was also carried out.
Reviewer 6 Report
The article "A blinded investigation: accentuated NK lymphocyte CD335 2 (NKp46) expression predicts pregnancy failures” describes the impact of NKp46 expression on NK cells on subsequent pregnancy outcomes in a population of naturally conceived women without a history of previous pregnancy complications.
The results presented herein are exciting and valuable. However, some obstacles must be overcome before the article is accepted for publication.
These, among others, include:
The synonyms like wg, y/o, GPL, PBS, etc., although self-explanatory to most practitioners in the corresponding clinical/research field, should be followed by their full names in parenthesis when mentioned for the first time in the manuscript text. Also, the authors should choose between aCL and ACL synonyms and use one of them throughout the text without variation.
Line 70… The Methods section should be renamed into - Patients and Methods- section. Also, the schematic illustration of the patient group's inclusion and exclusion criteria and the final number of patients included in the study would be helpful.
Line 92… The authors have written: “Patients was <36 y/o., have no history of repeated reproductive losses (not more than 1 previous pregnancy failure), autoimmune, chronic or acute infection. Anamnestic 93 and clinical data was shown in (Table1).” This sentence seems contradictory to the statement in lines 66-68: “In this part we study diagnostic prognostic value of NKp46 expression on NK cells for subsequent pregnancy outcome in natural conceived women without previously pregnancy complications.” The same applies to the sentence: “In this study, we investigated immunological parameters in early pregnant women without a burdened history and previous reproductive losses”(lines 261-263). The authors should reformulate these sentences or redefine the corresponding patient group.
Line 112… The authors have written: “Unfortunately, due to the war with the Muscovites, we were able to contact and obtain clinical information regarding 78 results of pregnancy ongoing only.” Although I do understand and humanly sympathize with the author's feelings regarding the unadjusted war (Russian Federation aggression) to which their country is unfortunately and unjustly subjected, nevertheless, I suggest the deletion of the term “with the Muscovites” and its replacement with the more appropriate term (e.g., ongoing war, or similar). The same applies to “war with orcs” in line 328. Again, this is a somewhat understandable term regarding the unjust Russian Federation aggression to Your country but still entirely inappropriate for a scientific journal of this kind.
In the section: “Assessment of NKp46 expression” the authors should provide the catalog numbers of antibodies used in the study.
Line 157… The authors should put the number 16 in parenthesis (if this is a reference number) or delete it from the sentence.
Line 174-177… The authors have written: “The statistical analysis of the results was performed using Fisher’s Exact Test (T, unpaired, non-parametric, two-sided P value ) and (95% Confidence Intervalusing the 175 approximation of Woolf) In Stat version 3.0 for Windows Graph Pad Software Inc., (San 176 Diego, CA, USA)”. The sentence should be reformulated. The text should also state the P value chosen to be statistically significant. Also, some parts of the manuscript text should be subjected to English language editing.
The results presented in the result section should be more precisely connected to corresponding figures and tables (i.e., corresponding figures and tables should be indicated in parenthesis in all cases when necessary). Also, the -*Significant difference compared to successful pregnancy outcomes and control groups- should be incorporated in a Figure 1 legend. The same applies to the corresponding statistically significant value designation in Figure 2.
Line 231… It seems that the term PF should not be in parenthesis.
Line 258-259… The authors have written: “The study of such groups of women could not exclude the influence of repeated miscarriage so n immunological parameters, in particular he expression of NKp46”. The sentence should be reformulated (n and he should be corrected and replaced). Also, the English language editing of the manuscript is suggested.
A major revision of the manuscript is suggested.
Author Response
The article "A blinded investigation: accentuated NK lymphocyte CD335 2 (NKp46) expression predicts pregnancy failures” describes the impact of NKp46 expression on NK cells on subsequent pregnancy outcomes in a population of naturally conceived women without a history of previous pregnancy complications.
The results presented herein are exciting and valuable. However, some obstacles must be overcome before the article is accepted for publication.
These, among others, include:
The synonyms like wg, y/o, GPL, PBS, etc., although self-explanatory to most practitioners in the corresponding clinical/research field, should be followed by their full names in parenthesis when mentioned for the first time in the manuscript text. Also, the authors should choose between aCL and ACL synonyms and use one of them throughout the text without variation.
-We correct it
Line 70… The Methods section should be renamed into - Patients and Methods- section.
-We correct it
Also, the schematic illustration of the patient group's inclusion and exclusion criteria and the final number of patients included in the study would be helpful.
-We add it in supplementary Tab1
Line 92… The authors have written: “Patients was <36 y/o., have no history of repeated reproductive losses (not more than 1 previous pregnancy failure), autoimmune, chronic or acute infection. Anamnestic 93 and clinical data was shown in (Table1).” This sentence seems contradictory to the statement in lines 66-68: “In this part we study diagnostic prognostic value of NKp46 expression on NK cells for subsequent pregnancy outcome in natural conceived women without previously pregnancy complications.” The same applies to the sentence: “In this study, we investigated immunological parameters in early pregnant women without a burdened history and previous reproductive losses”(lines 261-263). The authors should reformulate these sentences or redefine the corresponding patient group.
-We correct it
We plan to have a group without a burdened history and previous reproductive losses.
But in real life some doctors think that one pregnancy failure its not a “burdened history”
Line 112… The authors have written: “Unfortunately, due to the war with the Muscovites, we were able to contact and obtain clinical information regarding 78 results of pregnancy ongoing only.” Although I do understand and humanly sympathize with the author's feelings regarding the unadjusted war (russian federation aggression) to which their country is unfortunately and unjustly subjected, nevertheless, I suggest the deletion of the term “with the Muscovites” and its replacement with the more appropriate term (e.g., ongoing war, or similar). The same applies to “war with orcs” in line 328. Again, this is a somewhat understandable term regarding the unjust Russian Federation aggression to Your country but still entirely inappropriate for a scientific journal of this kind.
-We correct it
In the section: “Assessment of NKp46 expression” the authors should provide the catalog numbers of antibodies used in the study.
-We add it
Line 157… The authors should put the number 16 in parenthesis (if this is a reference number) or delete it from the sentence.
-We correct it
Line 174-177… The authors have written: “The statistical analysis of the results was performed using Fisher’s Exact Test (T, unpaired, non-parametric, two-sided P value ) and (95% Confidence Intervalusing the 175 approximation of Woolf) In Stat version 3.0 for Windows Graph Pad Software Inc., (San 176 Diego, CA, USA)”. The sentence should be reformulated. The text should also state the P value chosen to be statistically significant.
-We add it
Also, some parts of the manuscript text should be subjected to English language editing.
-We have made detailed proofreading of the English language and made numerous corrections.
The results presented in the result section should be more precisely connected to corresponding figures and tables (i.e., corresponding figures and tables should be indicated in parenthesis in all cases when necessary). Also, the -*Significant difference compared to successful pregnancy outcomes and control groups- should be incorporated in a Figure 1 legend. The same applies to the corresponding statistically significant value designation in Figure 2.
-We add it
Line 231… It seems that the term PF should not be in parenthesis.
-We correct it
Line 258-259… The authors have written: “The study of such groups of women could not exclude the influence of repeated miscarriage so n immunological parameters, in particular he expression of NKp46”. The sentence should be reformulated (n and he should be corrected and replaced). Also, the English language editing of the manuscript is suggested.
We correct it
Thank you for your attention to our article, advice and comments. We corrected the manuscript according to the comments. A detailed correction of the English language was also carried out.
Round 2
Reviewer 5 Report
Dear author’s
Thank you for your revised version.
We agree with you in term of the Russian agresion but in a scientific medical paper i think that all this information should be excluded.
Author Response
I have removed this title. I agree that it is probably not worth writing this title in a respectable journal)